# Revalidation of DNA Fragmentation Analyses for Human Sperm—Measurement Principles, Comparative Standards, Calibration Curve, Required Sensitivity, and Eligibility Criteria for Test Sperm

**DOI:** 10.3390/biology13070484

**Published:** 2024-06-28

**Authors:** Satoru Kaneko, Yuki Okada

**Affiliations:** 1Laboratory of Pathology and Development, Institute for Quantitative Biosciences, The University of Tokyo, 1-1-1 Yayoi, Bunkyo, Tokyo 113-0032, Japan; ytokada@iqb.u-tokyo.ac.jp; 2Sperm-Semen-Epididymis-Testis (SSET) Clinic, 1-5 Kanda-Iwamoto, Chiyoda, Tokyo 101-0033, Japan

**Keywords:** DNA fragmentation, single-cell pulsed-field gel electrophoresis, human sperm, sperm chromatin structure assay, sperm chromatin dispersion test, comet assay, quantitative analysis

## Abstract

**Simple Summary:**

Double-strand breaks (DSBs) in a single nucleus are usually measured using the sperm chromatin structure assay (SCSA), sperm chromatin dispersion (SCD) test, and comet assay (CA). Mono-dimensional single-cell pulsed-field gel electrophoresis (1D-SCPFGE) and angle-modulated two-dimensional single-cell pulsed-field gel electrophoresis (2D-SCPFGE) were developed to observe DNA fragmentation in separated motile sperm. Comparative standards, calibration curves, required sensitivity levels, and eligibility criteria for test sperm were set up to validate the measurement principles of these tests. The conventional methods overlooked the interference of nucleoproteins in their measurements. In-gel proteolysis improves the measurement accuracy levels of 1D- and 2D-SCPFGE. Naked DNA is suitable for comparative standards and test specimens. Moreover, several dysfunctions that might induce DNA damage are observed in the separated motile sperm. Overall, the discussion highlights the need to revisit the conventional univariable analyses based on the SCSA, SCD test, and CA. Human infertility is a complex syndrome, and the aim of quality control in intracytoplasmic sperm injection is to identify the underlying dysfunctions remaining in the separated motile sperm that render them ineligible for injection. Multivariable analyses with special consideration to confounding factors are necessary in future cohort studies.

**Abstract:**

(1) Background: Double-strand breaks (DSBs) in a single nucleus are usually measured using the sperm chromatin structure assay (SCSA), sperm chromatin dispersion (SCD) test, and comet assay (CA). Mono-dimensional single-cell pulsed-field gel electrophoresis (1D-SCPFGE) and angle-modulated two- dimensional single-cell pulsed-field gel electrophoresis (2D-SCPFGE) were developed to observe DNA fragmentation in separated motile sperm. (2) Methods: Comparative standards, calibration curves, required sensitivity levels, and eligibility criteria for test sperm were set up to validate the measurement principles of these tests. (3) Results: The conventional methods overlooked the interference of nucleoproteins in their measurements. In-gel proteolysis improves the measurement accuracies of 1D- and 2D-SCPFGE. Naked DNA is suitable for comparative standards and test specimens. Moreover, several dysfunctions that might induce DNA damage are observed in the separated motile sperm. Overall, the discussion highlights the need to revisit the conventional univariable analyses based on the SCSA, SCD test, and CA. (4) Conclusions: Human infertility is a complex syndrome, and the aim of quality control in intracytoplasmic sperm injection is to identify the underlying dysfunctions remaining in the separated motile sperm that render them ineligible for injection. Multivariable analyses with special consideration to confounding factors are necessary in future cohort studies.

## 1. Overview

The three major DNA-related contributors to human male infertility are de novo mutations, a decline in DNA repair capacity, and DNA fragmentation. Each of us is born with a few novel genetic changes, known as de novo mutations, that occur either during gamete formation or postzygotically [1]. Pathogenic de novo mutations that cause severe congenital anomalies are mostly of paternal origin [2], and they play a prominent role in severe male infertility [3]. Apoptosis regulates spermatogenesis in the testis during the spermatogonium, spermatocyte, and spermatid phases [4]. More than half of differentiating spermatogenic cells undergo apoptosis before maturing into sperm and are selectively and rapidly eliminated through phagocytosis by Sertoli cells [5]. Mature sperm lack DNA repair mechanisms and depend on maternal factors for repair after fertilization [6]. Due to the absence of homologous templates, zygotic repair of paternal double-strand breaks (DSBs) depends on non-homologous repair mechanisms that are considered to be error-prone. Moreover, apoptosis degenerates the sperm awaiting ejaculation in the epididymis [4,5]. DNA fragmentation is the most characteristic feature of the apoptotic process; upon activation, Ca/Mg-dependent endonucleases cleave DNA strands into nucleosomal subunits of ~185 base pairs in length [7]. Although intracytoplasmic sperm injection (ICSI) is a major fertilization procedure in assisted reproductive technology (ART), many cohort studies have revealed that DNA fragmentation in human sperm nuclei is a major risk factor, not only for early post-implantation embryo development [8], but also for sperm-derived congenital anomalies in ART [9].

For the past two decades, numerous researchers have discussed the significance of nonspecific single-nuclear DNA damage based on the sperm chromatin structure assay (SCSA) [10,11], sperm chromatin dispersion (SCD) test [12,13], and comet assay (CA) [14,15]. We have developed mono-dimensional single-cell pulsed-field gel electrophoresis (1D-SCPFGE) to observe single-nucleus DNA fragmentation [16,17,18]. Even in normozoospermic semen, this method has revealed that almost all of the immotile sperm in the ejaculate already exhibit end-stage DNA fragmentation [18,19,20]. Importantly, we discovered that the apparent density of motile sperm with fibrous DNA was 1.12–1.17 g/mL, whereas that of the immotile sperm exhibiting end-stage DNA fragmentation was higher than 1.17 g/mL [18,19,20]. The motile and immotile sperm were easily separated from each other using single-layer density-gradient centrifugation [18,19,20]. These two cell types differed in terms of not only the sequentiality of DNA [18,19,20] but also the integrity of the plasma and mitochondrial membranes [20], the status of the acrosome [20], the production of endogenous reactive oxygen species (ROS) in the midpiece [20], and antigenicity for anti-sperm antibodies [19]. These differences corresponded to those between the cells that had not yet undergone apoptosis and those that had. The difference in apparent density was due to a decrease in apoptotic volume [21].

As mature sperm lack DNA repair mechanisms [6], the critical threshold of DSBs in the nucleus is very low [22,23]. The incidence of sperm-derived congenital anomalies is not proportional to the number of DSBs, as a number exceeding the threshold results in fertilization failure or pregnancy loss. In conventional genome sequencing, DNA is extracted from a population of cells. Hence, genetic variations unique to individual cells are lost to the population average, and de novo mutations in a single cell are concealed in the bulk signal. Highly advanced technologies, e.g., high-throughput, single-nuclear, whole-genome amplification [24,25], are major candidates for use in detecting de novo mutations. However, even after the motile and immotile sperm have been separated, some of the motile sperm contain nonspecific DNA damage [16,17,18,19,20].

In the present review, we first introduce analytical methods to observe the early symptoms of DNA fragmentation in the motile sperm fraction, including the establishment of comparative standards, the calibration curve, the required sensitivity level, and the eligibility criteria.

Human infertility is a complex syndrome influenced by various dysfunctions besides DNA damage in the separated motile sperm fraction. Therefore, well-designed multivariate analyses with special consideration to confounding factors are necessary to conduct cohort studies. Subsequently, we discuss the design of cohort studies in which the pathology and epidemiology of DNA fragmentation are evaluated.

## 2. First Step of Qualitative-Method Validation—Establishment of Comparative Standards

We first purified the sperm with fibrous DNA, and those with granular segments from human semen; the detailed procedures for the separation of the two types of sperm have been described in our previous reports [18,19,20]. Briefly, the diluted semen was fractionated by means of the sedimentation equilibrium in an isotonic Optiprep (final apparent density of 1.17 g/mL, hereafter referred to as OP; Axis Shield, San Jose, CA, USA) and differential velocity sedimentation in an isotonic, 90% Percoll (GE Healthcare, Chicago, IL, USA) density gradient.

The sperm from the sediment of the OP/intermediate layer of the Percoll solution were almost immotile and auto-agglutinated, and their apparent density was estimated to exceed 1.17 g/mL. The fraction was termed “denatured sperm” (DS). The sperm recovered from the interface layer of OP/sediment of the Percoll solution was progressively motile, and their apparent density was estimated to be in the range of 1.12–1.17 g/mL. Finally, the motile sperm fraction was prepared using the swim-up method, and termed “live sperm” (LS).

Multi-step validations are necessary to ensure the proof of principle and the quantitative performance of DNA fragmentation analyses; they require the researcher to distinguish LS and DS, naturally occurring extremes, at the first-step qualitative-method validation.

## 3. 1D-SCPFGE

Traditional pulsed-field gel electrophoresis (PFGE) [26,27] is widely used for bacterial typing. Typically, cells are embedded in an agarose plug, proteolyzed, and further digested with restriction endonucleases [28]. Then DNA segments are separated in macro-gel to analyze the banding patterns. We developed micro-PFGE to observe single-nuclear DNA sequentiality without downsizing. The DNA morphology can be directly observed under a microscope using 1D-SCPFGE, as described in previous reports [18,19,20].

The in-gel-digested LS exserted the tips of its DNA fibers outward (Figure 1A). After 1D-SCPFGE, most of the LS exhibited a bundle of elongated long-chain fibers without any segment (Figure 1B). The remainder of the LS exhibited a few to several dozen fibrous segments beyond the anterior end of the elongated fibers. In contrast, in the DS, granular segments were circumferentially dispersed (Figure 1C), and upon 1D-SCPFGE, the segments resembled comet tails in the CA [14,15] (Figure 1D). Thus, according to the 1D-SCPFGE features, we define sperm with a bundle of elongated long-chain fibers without any fibrous segments as intact and those with at least one fibrous fragment as damaged.

As described in Section 4, we also developed a novel angle-modulated two-dimensional single-cell pulsed-field gel electrophoresis (2D-SCPFGE) [29]. This revealed some technical issues in 1D-SCPFGE; a fraction of long-chain fibers remained at the electrophoretic origin and long segments were tangled in the bundle of elongated fibers, even after the first run. Therefore, the definitions of intact and damaged obtained via 1D-SCPFGE are incomplete. Thus, 1D-SCPFGE can be used to observe mid- or end-stage DNA fragmentation, but not the early stages.

## 4. Conventional Analytical Methods for DNA Fragmentation Do Not Pass the First Step of Qualitative Validation

Studies comparing LS and DS have revealed various technical problems in the measurement principles of the SCSA [10,11], SCD test [12,13], and CA [14,15]. They do not pass the first step of qualitative validation.

### 4.1. Lack of Proteolysis Produces False Negative Results in the CA

The CA is used to estimate DNA damage based on the number of granular segments discharged from the origin, the so-called “comet tail” [14,15]. The 1D-SCPFGE visualized the progression of the fragmentation; at first, a few large, fibrous segments appear, and cleavage proceeds until all DNA fibers are degraded into granular segments [16,17,18,19,20]. Our observations raised doubts over whether the comet tail was derived from sperm with end-stage fragmentation [16,17,18]. Human sperm nucleoproteins include protamines [30,31], histones [32], condensins [33], and cohesins [33]. The DNA-nucleoprotein complex is fixed to the nuclear membrane with the nuclear matrix [34,35]. The CA extracted nucleoproteins, such as protamines [36], using a high-salt solution [14,15]. In-gel proteolysis of DS dramatically increases the number of granular segments extended via electrophoresis. In a neutral CA, unextracted nucleoproteins remain fixed to the DNA, and only a fraction of the granular segments unfixed were extended as the comet tail [18]. Naked chromosomal DNA fibers in LS are very susceptible to degradation via alkaline hydrolysis; a mere 10 mmol/L NaOH cleaves such fibers into granular segments [18]. Alkaline CA treated DNA with 300 mmol/L NaOH, while the binding capacities of the unextracted nucleoproteins tolerate for this radical treatment, they still fixed to the DNA, preventing migration of the newly generated granular segments [18]. Therefore, a lack of proteolysis produces false-negative results in both neutral and alkaline CAs. Furthermore, the CA without a pulsed-field current is insufficient to extend the fibrous DNA [14,15].

The enzymatic properties of bovine pancreatic trypsin (EC.3.4.21.4) [37], with respect to substrate specificity and pH dependency, are optimal for the digestion of sperm embedded in agarose. Protamines are arginine-rich, basic proteins in which the guanidyl residue is tightly coupled with phosphoric acid in DNA through electrostatic bonding, and neighboring protamines are cross-linked through disulfide bonds [30,31,36]. Trypsin specifically cleaves the carboxyl ends of lysine and arginine residues [37]. In-gel digestion of the embedded cells needs to be initiated after the agarose is completely solidified to avoid free diffusion of the DNA fibers. As the activity of trypsin is strictly dependent on pH, trypsin can be kept inactive at a pH of 4.7 until the gel solidifies and then reactivated by immersing the gel into the cell-lytic reagents (pH = 8). As commercial preparations of trypsin are usually contaminated with pancreatic deoxyribonucleases (DNases), they are not suitable for DNA studies as provided. To remove autolyzed trypsin and DNases, twice-crystallized bovine pancreatic trypsin must be further purified via affinity chromatography by using lima bean trypsin inhibitor–conjugated Sephacryl [38]. The purified trypsin should be stored in a solution with a pH < 2.0 to avoid autolysis. Trypsin undoubtedly has a high level of performance, but lacks versatility.

Proteinase K [39] has a chymotrypsin-like broad substrate specificity for aliphatic and aromatic residues of amino acids, and acts in a wide pH range; ready-to-use preparations are supplied commercially for the extraction of DNA from somatic cells. However, proteinase K is not very effective against protamines. Competitive dissociation of the protamine–DNA complexes with sodium dodecyl sulfate (SDS) and subsequent digestion of other nucleoproteins with proteinase K allows migration of DNA fibers via 1D-SCPFGE [29].

### 4.2. Red Fluorescence in the SCSA Is Derived from Nucleoproteins but Not from Single-Stranded DNA

The SCSA employs a simple bisection principle wherein the intercalation of monomeric acridine orange (AO) into double-stranded DNA or the adsorption of oligomeric AO to single-stranded DNA produces green or red fluorescence, respectively [10,11]. After AO staining, the nucleus and cytoplasm of lymphocytes yield green and red fluorescence, respectively, and in-gel tryptic digestion of the sperm caused the red fluorescence to vanish [40]. These phenomena raise doubts regarding whether the red fluorescence was truly derived from the single-stranded DNA.

The SCSA is routinely performed using flow cytometry [10,11]. The fluorescent profiles of LS and those that underwent radical treatment (96 °C for 20 min) to destroy the DNA structure yielded similar cytograms. DNA and nucleoproteins are packed rigidly in the sperm head, and the combination of the two colors leads to color variation [40]. Prolonged exposure under a fluorescence microscope causes photobleaching of fluorescent dyes and the concurrent photo-breakage of DNA. Under such conditions, sperm nuclei do not undergo simple color quenching but rather a progressive, time-dependent discoloration from red to green, which contradicts the principle of the SCSA [40]. The red fluorescence is derived from the AO adsorbing to nucleoproteins rather than the single-stranded DNA. Discoloration during prolonged exposure is due to the higher tolerance of intercalated AO compared to that of AO adsorbed to proteins. The SCSA is normally used to examine unseparated semen, and a DNA fragmentation index (DFI) is calculated from the distribution of signals on the cytogram [10,11]. With such an indirect signal analysis, one must carefully interpret the origin of the signals on the cytogram. One would have to account for signals from non-sperm debris to calculate an accurate DFI.

Several researchers have referred to the interaction of monomeric and oligomeric AO with non-DNA materials [41,42,43,44,45]. AO can permeate organelle membranes, and proton pump-driven intra-lysosomal acidity facilitates the accumulation of permeated AO, the oligomeric aggregates of which exhibit a red shift. When the lysosomal pH rises due to membrane damage, oligomeric AO dissociates to form monomers, which shifts the emission from orange to green. Thus, AO acts as an indicator of lysosomal localization and proton pump damage. The principle of SCSA misunderstands the chemical properties of AO; the latter adsorbs to various intercellular materials besides DNA [40].

### 4.3. The Halo in the SCD Test Comprises Unextracted Nucleoproteins Adhered to DNA Fibers

After in-gel high-salt extraction of nucleoproteins in human sperm, crystal violet (CV) staining is used to visualize the DNA fibers that are radiated outward. In the SCD test, the extent of DNA damage is defined as being inversely proportional to the area of the violet halo [12,13]. As the violet halo disappears upon in-gel tryptic digestion, CV staining visualizes a real picture of unextracted nucleoproteins adhered to DNA fibers [18]. The SCD test could not clearly distinguish the halos of LS and DS [18]. The SCD test overlooked the interference of nucleoproteins in their measurement.

### 4.4. Tightly Packed DNA-Nucleoprotein Complex Physically Blocks Permeation of Certain Dyes and Terminal Deoxynucleotidyl Transferase

After reduction of inter- and intramolecular disulfide cross-linking of protamine with dithiothreitol (DTT), guanidine sulfate competitively dissociates the DNA-nucleoprotein complex, which results in sperm head swelling [20,46]. The dye exclusion assay with trypan blue, which is widely used to observe the plasma membrane integrity of somatic cells, resulted in a lack of staining for almost all the membrane-excluded LS (Figure 2A), whereas the swollen head was stained deep blue [20] (Figure 2B). A similar phenomenon has been observed with bromophenol blue and ponceau 4R [20]. The tightly packed DNA-nucleoprotein complex physically blocks the permeation of certain organic dyes.

The terminal deoxynucleotidyl transferase (TdT)-mediated dUTP nick-end labeling (TUNEL) assay [47] catalyzes the addition of fluorescent-dUTP at the 3′-OH end of damaged DNA segments. The assay detects the nicks in the non-swollen DS (Figure 2C) and the swelling increases the intensity of the fluorescence (Figure 2D). The tightly packed nucleus physically blocks permeation of TdT, which has a molecular weight of 32,360 [48]. The signals observed in Figure 2C may be derived from the DNA segments at the surface. Pre-swelling of the sperm head is recommended for the TUNEL assay; however, its sensitivity may be insufficient to detect the early stage of fragmentation.

## 5. Angle-Modulated 2D-SCPPFGE

The 2D-SCPFGE sleaves the tangled mass of DNA fibers during the first short run, and the subsequent change in the electrophoretic direction causes them to elongate and align. Variations in the rotation angle and the current-application time provide diverse alignment profiles (Figure 3) [29]. The main aim of this method is to detect small numbers of fibrous DNA segments that appear in the early-stage fragmentation. Figure 3A–C summarize the progress of fragmentation. No segments were found in the inner angle of the fan-like shape formed by the elongated DNA fibers (Figure 3A). A few fibrous segments appeared in the inner angle of the fan (Figure 3B). As fragmentation proceeds, fibrous fragments of varying sizes are assembled in the inner angle of the fan (Figure 3C). Figure 3D shows the electrophoretic condition used to align a set of single-nuclear DNA fibers in parallel without overlapping; they followed S-shaped curves according to the switching intervals of the current.

The 2D-SCPFGE revealed technical issues with 1D-SCPFGE. The lack of a change in direction renders 1D-SCPFGE incapable of drawing out the long fibrous segments from the bundle of fibers [29]. Theoretically, the longest possible segments would arise from chromosome 1 being cleaved in half-and-half, and although 2D-SCPFGE with angle modulation has an improved sensitivity, compared to 1D-SCPFGE, whether it can draw out such long segments remains unclear. An experiment should be performed in which a few to several dozen cleavages are made in a nucleus to know the detectable upper sizes limit. A set of size markers will also be needed to calibrate the segments observed in the micrographs.

LS and DS, naturally occurring extremes, are suitable for qualitative-method validation. The calibration standard required to determine the limit of sensitivity must have a small number of cleavages and should be prepared via artificial cleavage of LS. To date, we have examined the cleavage of DNA fibers by means of heat denaturation [16,40], alkaline hydrolysis [18], hydroxy radicals produced via the Fenton reaction [49], and restriction endonuclease digestion (EcoR1) [29]. In each case, 1D-SCPFGE confirmed that the DNA was cleaved in a dose-dependent manner; the dose-cleavage curves were, however, too steep to simulate early-stage fragmentation.

One task to be explored in the future using 2D-SCPFGE is determining from which chromosomes the fibers are derived. Fluorescence in situ hybridization (FISH) enables mapping of a specific DNA sequence in metaphase chromosomes and interphase nuclei. FISH applied to naked DNA fibers, the so-called “DNA fiber-FISH”, yields the highest mapping resolution [50]. The extended long DNA fibers are very fragile in an aqueous solution, and the oscillation of the fibers due to micro-Brownian motion readily cleaves the DNA strands (personal observation). After the fibers are aligned by means of 2D-SCPFGE, one technical issue that remains to be solved is the chemical fixation of the extended long DNA fibers on the solid phase. The 2D-SCPFGE is a pioneering approach and is presently the most sensitive method used to image large DNA. Therefore, no precedents exist for the preparation of standards with the lengths associated with chromosomal-level DNA fibers. Until calibration standards are established, the quantitative performance of 2D-SCPFGE will remain undefined. We tentatively propose 2D-SCPFGE as a reasonably accurate method used to determine whether DNA is sequentially intact, which is defined as the absence of long DNA segments in the region of interest.

## 6. How to Develop and Validate Analytical Methods for DNA Fragmentation

We learned six lessons from the conventional analytical methods. First, with regard to comparative standards, chemical probes often interact with various intercellular materials besides DNA; therefore, multi-step validations using qualitative and quantitative standards, and subsequent cross-validations among different instruments, are essential to verify the measurement principles. For example, observations using flow cytometry, SCPFGE with in-gel proteolysis, and fluorescence microscope revealed that the red fluorescence in SCSA is derived from AO adsorbing to nucleoproteins rather than single-stranded DNA [40]. Second, as the critical threshold of DSBs in a sperm is extremely low, analytical methods need to have sufficient sensitivity to detect a low level of cleavage in sperm in order to predict sperm-derived congenital anomalies. Third, the calibration curves are essential in ensuring the quantitative performance. Dose-dependent artificial DNA cleavage is suitable for making a set of the calibration standards; as aforementioned, those that mimic the early-stage level are not established. Prokaryotic DNA without downsizing may be potential candidates for calibration standards used to measure the length of DNA. Fourth, the eligibility criteria for test samples should be rigorously defined and should be limited to the separated motile-sperm fraction. The 1D-SCPFGE originally suggested that almost all immotile sperm are in end-stage fragmentation [19,20]. However, once the cells are fixed and stained or embedded and lysed, determining the type of sperm from which the DNA is derived is difficult. Fifth, when analytical technologies normally used on somatic cells are imported for use on sperm, the unique cellular features of sperm should be considered. For instance, the tightly packed DNA-nucleoprotein complex physically blocks permeation of TdT (Figure 2), and enzymatic analyses, including TUNEL assays, are impeded due to steric hindrance. Sixth, the CA, SCSA, and SCD test overlooked the interference of nucleoproteins in their measurement principles. When compared with the neutral CA, in-gel proteolysis and subsequent SCPFGE improve the quantitative performance in DNA fragmentation analyses (Figure 1 and Figure 2); the naked DNA mass should be used as the comparative standard and the test specimen.

## 7. Separated Motile Sperm Include Various Impairments Besides DNA Fragmentation

For many years, sperm concentration, sperm motility, and the morphology of the head have been assessed as key parameters of “semen quality”, a term that is used as a surrogate for male fecundity. In ICSI, intra-operative sperm selection usually depends on gross morphology and motility. However, as cited above, 1D-SCPFGE revealed that even after separation of the motile sperm, some exhibit fibrous DNA segments [16,17]. Therefore, we developed several pre-operative methods used to evaluate sperm quality, such as 2D-SCPFGE for the detection of the early symptoms of DNA fragmentation [29], sperm-specific dye- and lectin-exclusion assays for assessment of the plasma and acrosomal membranes [20], the dye-retention assay for assessment of the mitochondrial organelle membrane, assessments of endogenous ROS in the mitochondria [20], and visualization of vacuoles in the sperm head [46,51]. These inspections highlight the importance of a multifaceted approach to the discussion of DNA fragmentation.

### 7.1. Sperm-Specific Two-Step Dye-Exclusion Assay to Observe Damage of Plasma Membrane

All of the organelles in somatic cells are grouped into one compartment and enveloped within the plasma membrane. Mammalian sperm are well known to differ greatly from somatic cells in terms of their membrane organization. The former is compartmentalized into at least four regions: the acrosomal cap at the anterior end of the head, the posterior region of the head, the midpiece, and the principal and terminal pieces of the tail. The acrosomal cap is composed of the plasma membrane and outer- and inner-acrosomal membranes [52,53]. In contrast, the posterior region of the head is enveloped solely within the plasma membrane, and the lack of cytoplasm places the nucleus just below the membrane. Thus, damage to the plasma membrane in this region may directly affect DNA integrity.

Ascorbic acid (AA) is a well-known antioxidant that acts as a pro-oxidant in the presence of transitional metals via the Fenton reaction [54]. We previously observed that AA cleaved DNA double strands in membrane-excluded human sperm [49], and that ethylene diamine tetra-acetic acid (EDTA) inhibits this action. DS with damaged plasma membranes have granular DNA segments [20]. Therefore, we hypothesized that damage to the plasma membrane will cause AA in the seminal plasma [55] to come into contact with DNA–transition metal complexes in the nucleus, causing DNA fragmentation.

We developed a sperm-specific two-step dye-exclusion assay with a high affinity for sperm-specific protamines (Figure 4) [20]. Reactive red 195 (RR195) and reactive blue 222 (RB222) both contain 5–6 sulfate residues per molecule, which strongly bind with guanidyl residues in protamines. The sperm were incubated with isotonic RR195, and the reaction mixture was adhered to a plane glass slide, treated with methanol to exclude the plasma membrane, and treated with RB222 for counter-staining. Heads that were stained red indicated that RR195 permeated through a damaged plasma membrane. Those that were stained blue indicated that the counter-dye permeated after membrane exclusion, suggesting that the plasma membranes had been intact. As shown in Figure 4, in contrast to LS, almost all of the DS had already undergone plasma-membrane impairment.

### 7.2. Observation of the Plasma and Acrosomal Membranes via Two-Step Concanavalin A-Labeling

The acrosomal region is covered with three layers: the plasma, outer-acrosomal membrane, and inner-acrosomal membrane [52]. During the acrosome reaction, the plasma and outer-acrosomal membranes fuse [53]; consequently, the DNA-nucleoprotein complex just below the acrosome is protected solely by the inner-acrosomal and nuclear membranes. Whether they block extracellular materials in the same way as the plasma membrane is unclear. Furthermore, no methods have been developed to detect damage to the inner-acrosomal membrane.

When Cy3-conjugated concanavalin A (Cy3-con A; Molecular Probes, Eugene, OR, USA) permeates through damaged plasma and outer-acrosomal membranes, it binds with high-mannose glycoproteins on the inner-acrosomal membrane. In one of our previous studies [20], sperm were incubated with isotonic Cy3-con A in the presence of methyl α-D-mannopyranoside, an antagonist of concanavalin A [56]. The reaction mixture was treated with methanol to exclude the plasma and outer-acrosomal membranes and subsequently incubated with Alexa 488-conjugated concanavalin A (Alexa488-con A; Molecular Probes), in the same manner as with Cy3-con A. Thereafter, the same field of view was observed under a fluorescent microscope, first with a green filter and then with a red filter. These images were digitally merged. Red fluorescence on the merged photograph indicated that the plasma and outer-acrosomal membranes had already been damaged, facilitating the permeation of Cy3-con A. Green fluorescence indicated that Alexa488-con A had bound to the inner-acrosomal membrane that was exposed after methanol treatment. As shown in Figure 5, in contrast to LS, almost all DS had damaged plasma and outer-acrosomal membranes.

### 7.3. Dye-Retention Assay for Endogenous ROS Generated in the Mitochondria

Live mitochondria generate endogenous ROS as metabolites of oxidative phosphorylation in the tricarboxylic acid cycle. CellROX Orange (Thermo Fisher Scientific, Waltham, MA, USA) permeates the mitochondria and produces orange fluorescence upon oxidation by ROS (Figure 6A) [57]. MitoTracker FM (Thermo Fisher Scientific), which permeates and is retained by the mitochondria regardless of the membrane potential of the organelle [58], produces green fluorescence in the midpiece (Figure 6B). It is employed in a dye-retention assay to observe the integrity of the mitochondrial membrane. The LS exhibit dual fluorescence derived from cellROX and MitoTracker FM, but both are not seen in DS [20]. These assays are the referential index of oxidative phosphorylation taking place in mitochondria with an intact organelle membrane.

### 7.4. Observation of Vacuoles in the Heads of Sperm

When we previously investigated ATP-activated channels in human sperm [46,51], we observed that a well-known, highly potent P2Y purinergic-receptor antagonist, reactive blue 2 (RB2) [59], strongly binds with protamine. It faintly stains the sperm head, which is observed as having a translucent bluish body, and the vacuoles appear as toneless spots. Staining with RB2, which has three sulfate residues, leads to the swelling and re-formation of the head into an oval shape and the disappearance of the vacuoles, regardless of their original features [46], just as seen with guanidine sulfate (Figure 2B) [20]. Comparison of RB2 staining with and without DTT is the referential index of local failure of inter- and intra-disulfide cross-linkage during spermiogenesis (Figure 7).

## 8. Multivariate Analyses of DNA Damage and Confounding Factors in Cohort Studies

As non-homologous zygotic repair of paternal DSBs is prone to error, de novo mutations in the zygote are mostly of paternal origin [2]. Single-strand breaks (SSBs) are one of the most frequent DNA lesions; they are produced daily in the nucleus during DNA repair. Base excision repair [60] is a primary response pathway for the repair of deleterious DNA lesions, including non-bulky DNA adducts, apurinic/apyrimidinic (AP) sites [61,62], and SSBs. If unrepaired or incorrectly repaired, these lesions threaten genetic integrity through their potential conversion to lethal DSBs during DNA replication. Single-nuclear DSBs and SSBs (AP sites) should preferably be analyzed together, although, considering their relative frequency, the latter may pose a greater risk to DNA integrity.

Early apoptotic events or oxidative stress induce phospholipid flip-flop in the plasma membrane [63]. Endogenous ROS in the mitochondria [20] and exogenous ROS due to plasma membrane damage at the posterior region of the sperm head [49] should be taken into account as potential confounding factors. Researchers have reported that protamine deficiency is responsible for vacuole formation, and, secondarily, for DNA fragmentation [64,65]. We have also observed that local failure of disulfide cross-linkage may play a critical role in determining sperm-head morphology, as well as vacuole formation [46]. The underlying mechanism might be either that insufficient cross-linkage destabilizes all of the DNA or that it damages only the DNA at the inner surface of the vacuole. Detection of the latter, however, is challenging at present, due to the difficulties involved in using probes with high molecular weight to permeate through the tightly packed nucleus. The age-dependent decline in embryonic repair of paternal DNA [60] is another potentially confounding factor.

In general, laboratory mice are generated via repeated inbreeding and natural delivery, and they remain highly fecund over many generations. In livestock farming, the seed bulls selected through progeny testing have high fecundity. The animal experiments using inbreeding mice or seed bull fit to human couples who pregnant spontaneously, in contrast, often un-fit to the infertile patients. Mice used as pathogenic animal models are classically generated through selective pedigree breeding; however, an animal model of male infertility cannot be sustainably propagated in such a way. Male infertility is not an issue in livestock farming. In contrast, therapy for male infertility is in high demand among monogamous humans.

Recent progress in gene manipulation has led to the creation of various knockout mice with particular gene variants. The diverse symptoms associated with human infertility are caused by interactions between congenital and acquired reproductive dysfunctions in both the man and the woman. Determining the genetic origins of the varying symptoms of infertile men may be difficult with current techniques. Thus, well-designed cohort studies with special consideration to confounding factors may accelerate our understanding of the etiology of male infertility.

## 9. Vision for the Future

As far as in the scope of the present review, the sperm in clinical ICSI should be selected according to the strict criteria: a motile sperm with normal head, midpiece, and tail morphologies; intact acrosomal and mitochondrial membranes; physiological oxidative phosphorylation in the mitochondria; and an absence of vacuoles and DNA damage such as DSBs and SSBs (AP sites). The fundamental framework involved in quality control for ICSI is to clarify which of the underlying dysfunctions in the motile sperm render them ineligible for injection. According to our preliminary observations, the infertile men who exhibited normal “semen quality under phase contrast microscopy” but suffered from repetitive ICSI failures exhibited a higher frequency of early-stage fragmentation, as well as vacuoles and damage to the plasma membrane, than did the men in couples who became pregnant within a few ICSI treatment cycles. The present review highlights the need to revisit some cohort studies based on the CA, SCSA, SCD test, and TUNEL assay. We are developing a modified 2D-SCPGFE, along with the corresponding calibration standards, for the simultaneous observation of early-stage DSBs and SSBs (AP sites). When this development is complete, we plan to conduct multivariate analyses with the novel method to determine the etiological significance of DNA damage in male infertility. Endogenous and exogenous ROS, types of plasma membrane damage, and formation of vacuoles should be taken into account as potential confounding factors; multifaceted approaches are necessary to find answers through cohort studies.

Another issue in ICSI quality control is intra-operative, non-invasive testing. The RR195-exclusion assay without counterstaining serves as an example of such a test; motile sperm that stain red should be discarded. Our preliminary experiments suggest that various acidic dyes, including those that contain sulfate residues, have high affinity to protamines [20]. We hope to discover a dye that is excluded completely by the intact plasma membrane; this would require detailed exposure experiments according to reproductive toxicity guidelines prior to clinical use.

## Figures and Tables

**Figure 1 biology-13-00484-f001:**
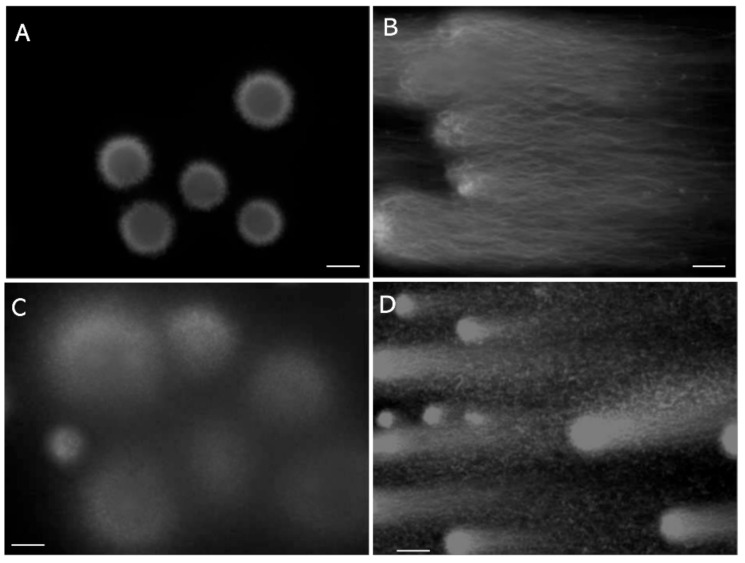
Mono-dimensional single-cell pulsed-field gel electrophoresis. (**A**): LS digested in the gel with trypsin. (**B**): Electrophoretogram of LS. (**C**): DS digested in the gel with trypsin. (**D**): Electrophoretogram of DS. The scale bars represent 20 μm.

**Figure 2 biology-13-00484-f002:**
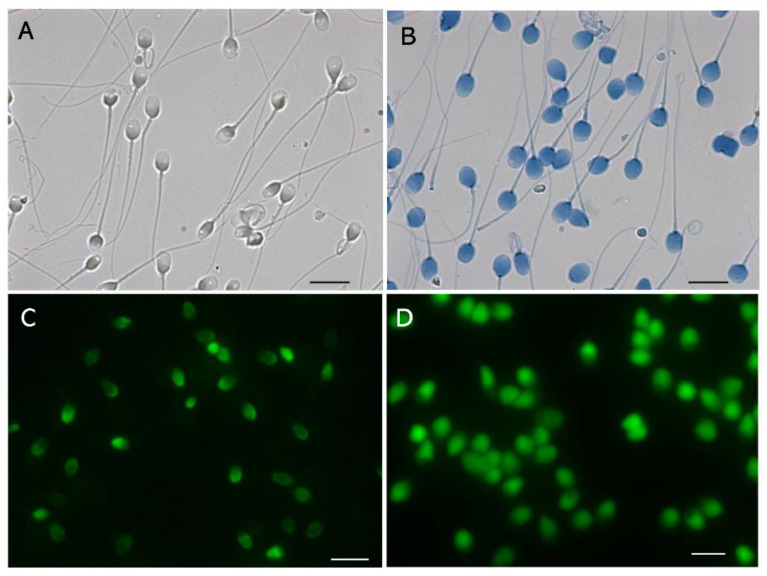
Tightly packed DNA-nucleoprotein complex physically blocks permeation of certain organic dyes. (**A**): Membrane-excluded LS stained with 0.05% trypan blue. (**B**): Membrane-excluded LS swollen with 0.1 mol/L guanidine sulfate and 5.0 mmol/L DTT, stained in the same manner as those in (**A**). (**C**): TUNEL assay for non-swollen DS, performed using the TUNEL Assay Apoptosis Detection Kit (Cosmo-Bio, Tokyo, Japan). (**D**): TUNEL assay for swollen DS, performed using the same kit. The scale bars represent 10 µm.

**Figure 3 biology-13-00484-f003:**
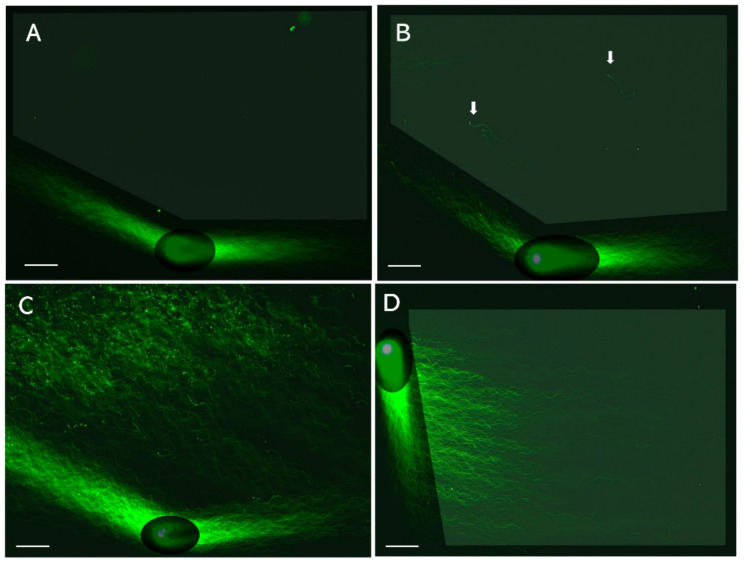
Angle-modulated two-dimensional single-cell pulsed-field gel electrophoresis. LS were electrophoresed by means of 2D-SCPFGE; for details of image enhancement, please see Reference [29]. (**A**): After the first run, the glass slide is turned 150 degrees (2D-SCPFGE-0-150, first run: 5.0 min; second run: 3.5 min). DNA fibers are elongated in a fan-like shape, and no DNA is observed in the inner angle of the fan. (**B**): The arrows indicate a few fibrous segments separated in the inner angle of the fan. (**C**): Segments of various sizes are assembled in the inner angle of the fan. (**D**): 2D-SCPFGE-(-75)-0 (3.0/10 min) aligned a set of DNA fibers in parallel without the need for downsizing. The scale bars represent 50 µm.

**Figure 4 biology-13-00484-f004:**
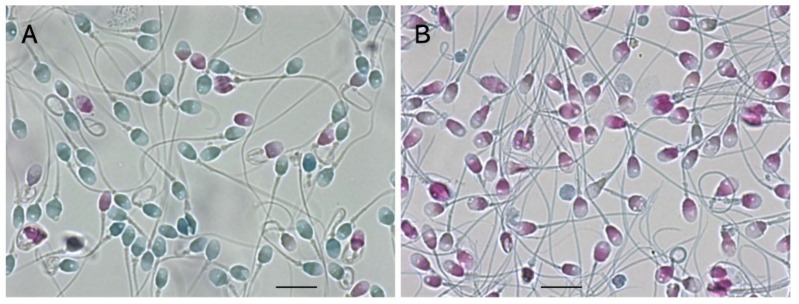
Sperm-specific two-step dye-exclusion assay for living sperm and denatured sperm. (**A**): LS. (**B**): DS. Scale bars represent 10 µm.

**Figure 5 biology-13-00484-f005:**
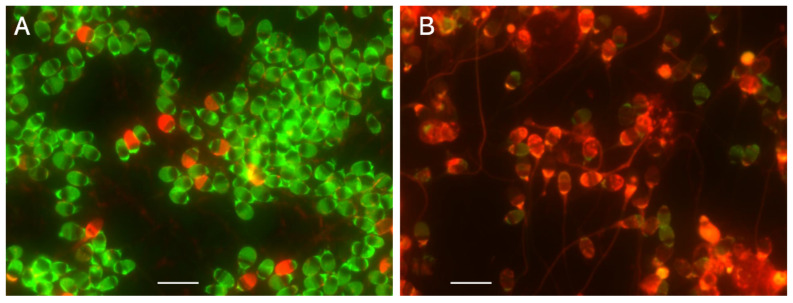
Sperm-specific two-step concanavalin-A labeling for living and denatured sperm. (**A**): LS. (**B**): DS. The scale bars represent 10 µm.

**Figure 6 biology-13-00484-f006:**
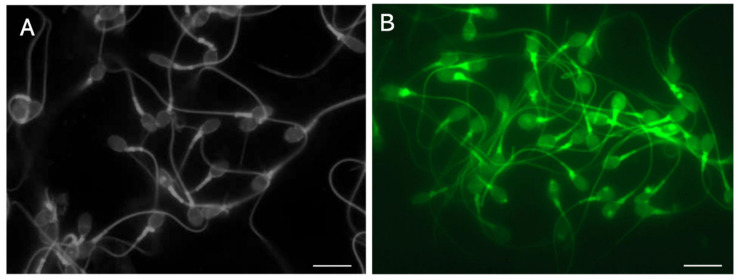
Dye-retention assays for mitochondria of living sperm. (**A**): The orange fluorescence was due to oxidative phosphorylation in the mitochondria of LS. The black-and-white photograph was taken with a highly sensitive charge-coupled device camera. (**B**): Green fluorescence was derived from the retention of MitoTracker FM in the mitochondria of LS. The scale bars represent 10 µm.

**Figure 7 biology-13-00484-f007:**
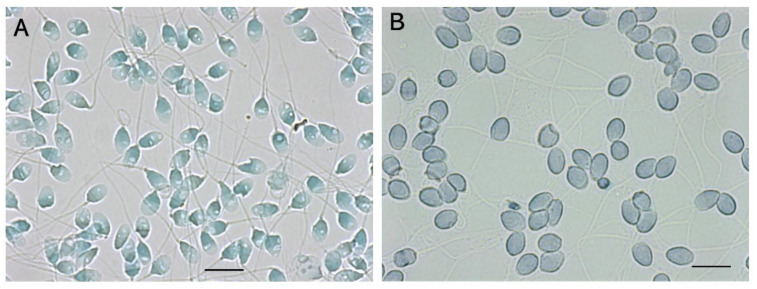
Observation of the vacuoles in human sperm with reactive blue 2 and swelling of the sperm head via reduction of disulfide bonds; LS were stained with 0.02% RB2 (0.1 mol/L Na_2_CO_3_–NaHCO_3_, pH = 10.0) in the absence (**A**) or presence (**B**) of 5.0 mmoL/L DTT. (**A**): The vacuoles are visualized as toneless spots. (**B**): DTT swelled the head and resulted in the disappearance of the vacuoles. The scale bars represent 10 µm.

## Data Availability

The datasets generated and analyzed in the present study are available from the corresponding author upon reasonable request.

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
