# Peer review of "Revalidation of DNA Fragmentation Analyses for Human Sperm—Measurement Principles, Comparative Standards, Calibration Curve, Required Sensitivity, and Eligibility Criteria for Test Sperm"

_biology, 2024, doi:10.3390/biology13070484_

Round 1
Reviewer 1 Report
Comments and Suggestions for Authors
This manuscript (ID Biology-3046274) entitled, “Revalidation of DNA fragmentation analyses for human sperm – measurement principles, comparative standards, calibration curve, required sensitivity, and eligibility criteria for test sperm – a step-by-step approach to cohort studies with multivariable analyses” is a review article where the authors Satoru Kaneko and Yuki Okada, introduce analytical methods to observe the early symptoms of sperm DNA fragmentation including the establishment of comparative standards, the calibration curve and the sensitivity requirement. The authors also discuss the design of cohort studies for evaluating the pathology and epidemiology of sperm DNA fragmentation. Also, the authors developed mono-dimensional single-cell pulsed-field gel electrophoresis (1D-SCPFGE) and angle-modulated two-dimensional single-cell pulsed-field gel electrophoresis (2D-SCPFGE) to observe sperm DNA fragmentation.
It is a well written review. The authors have discussed many aspects (some new) on this important topic of sperm DNA fragmentation. The following points/issues require further clarifications:
1) Lines 91-96: The discussion on the critical threshold of DSBs and its impact on fertilization and pregnancy is important.
2) Lines 111-120: A brief rationale for why these two extremes need to be distinguished in qualitative validation. A summary or highlight of these key steps help the reader to better understand without needing to refer it back and forth.
3) Lines 311-318: Need to emphasize practical aspects such as specific examples of qualitative and quantitative standards used in validation. How cross-validation among different instruments is conducted?
4) Lines 311-346: Highlight any novel aspects or innovations of each method compared to conventional approaches, emphasizing how they contribute to advancing the understanding of sperm function and fertility potential. Also, discuss the potential clinical implications of using these advanced assays in reproductive medicine or infertility treatment.
5) Lines 446-447: The sentence has grammatical mistakes and seems to be incomplete or unclear.
Comments on the Quality of English LanguageNo comments
Author Response
Dear reviewer 1: According to the suggestions by the reviewers, some sentences were inserted or replaced; the blue sentences: reviewer 1, and the red sentences: reviewer 2.
- Lines 91-96: The discussion on the critical threshold of DSBs and its impact on fertilization and pregnancy is important.
Line 84-85 “the critical threshold of DSBs in the nucleus is very low [22, 23].” We represented qualitatively as “low”, to determine the threshold, it is necessary to establish highly sensitive SCPFGE and the calibration standards. The quantitative measurement of “the threshold” is the final objective. The present review represents the value as “low”.
- Lines 111-120: A brief rationale for why these two extremes need to be distinguished in qualitative validation. A summary or highlight of these key steps help the reader to better understand without needing to refer it back and forth.
Line 105-121 According to the suggestion, the original sentences were replaced to the following sentences.
We, first, purified the sperm with fibrous DNA and those with granular segments from human semen, the detailed procedures for the separation of the two types of sperm were described in our previous reports [18-20]. Briefly, the diluted semen was fractionated by means of the sedimentation equilibrium in isotonic Optiprep (final apparent density of 1.17 g/mL, hereafter referred to as OP; Axis Shield, San Jose, CA, USA) and differential velocity sedimentation in an isotonic, 90% Percoll (GE Healthcare, Chicago, IL, USA) density gradient.
The sperm from the sediment of the OP/intermediate layer of the Percoll solution were almost immotile and auto-agglutinated, and their apparent density was estimated to exceed 1.17 g/mL. The fraction was termed “denatured sperm” (DS). The sperm recovered from the interface layer of OP/sediment of the Percoll solution was progressively motile, and their apparent density was estimated to be in the range of 1.12–1.17 g/mL. Finally, the motile sperm fraction was prepared using the swim-up method and termed “live sperm” (LS).
Multi-step validations are necessary to ensure the proof of principle and the quantitative performance of DNA fragmentation analyses, they require to distinguish LS and DS, naturally occurring extremes, at the first step qualitative-method validation.
- Lines 311-318: Need to emphasize practical aspects such as specific examples of qualitative and quantitative standards used in validation. How cross-validation among different instruments is conducted?
According to the suggestion, following sentences are inserted.
Line 323-326 For example, observation among flow cytometry, SCPFGE with in-gel proteolysis, and fluorescence microscope revealed that the red fluorescence in SCSA is derived from AO adsorbing to nucleoproteins rather than single-stranded DNA.
Line 331-332 Prokaryotic DNA without downsizing may be potential candidates for the calibration standards to quantify the fibrous DNA.
- Lines 311-346: Highlight any novel aspects or innovations of each method compared to conventional approaches, emphasizing how they contribute to advancing the understanding of sperm function and fertility potential. Also, discuss the potential clinical implications of using these advanced assays in reproductive medicine or infertility treatment.
According to the suggestion, following sentences are inserted.
Line 342-345 in-gel proteolysis and subsequent SCPFGE improve the quantitative performance in DNA fragmentation analyses (Fig. 1 and 2), the naked DNA mass should be used as the comparative standard and the test specimen.
- Lines 446-447: The sentence has grammatical mistakes and seems to be incomplete or unclear.
The sentence was corrected.
Line 471 In livestock farming, the seed bull selected through progeny testing have high fecundity.

Reviewer 2 Report
Comments and Suggestions for Authors
The topic chosen for this review is relevant and highlighting the importance of multifaceted approach to discuss DNA fragmentation is a contribution to the scientific community and the literature. I have some suggestions for the authors:
-The title is long and not clear.
-On the first part of the manuscript (including the abstract) there is an extensive use of the word "we", which does not fit with the writing of a review article.
-Infertility is not a symptom. ESHRE's definition of infertility is: “Disease characterized by the failure to establish a clinical pregnancy after 12 months of regular, unprotected sexual intercourse or due to an impairment of a person's capacity to reproduce, either as an individual or with their partner”
-I would highly recommend editing on the organization of the manuscript. It starts by describing gold standard protocols for sperm selection such as density gradient and swim-up and than passing to the work that the authors have done. In my opinion the chapter 1 can be deleted from the manuscript.
-The enumeration is not correct: Chapter 1 is followed by the chapter 2.1 and then proceeds with the chapter 3.
-The last part of the article is written as an original article in which the authors highlight that they are working on "developing of a modified 2D-SCPGFE 471 and the corresponding calibration standards for the simultaneous observation of early-stage DSBs and SSBs (AP sites)". This is confusing for the reader as i would recommend the authors to explain and highlight once again the pros and the cons of the protocols described in the manuscript.
Comments on the Quality of English LanguageThe authors used academic writing and there were not any mistakes detected. I would suggest the authors to work on the clarity of the text. Some paragraphs can be shorter and clearer.
Author Response
Dear reviewer 2: According to the suggestions by the reviewers, some sentences were inserted or replaced; the blue sentences: reviewer 1, and the red sentences: reviewer 2.
-The title is long and not clear.
Line 1-4: The title is shortened according to the suggestion. “Revalidation of DNA fragmentation analyses for human sperm – measurement principles, comparative standards, calibration curve, required sensitivity, and eligibility criteria for test sperm”.
-On the first part of the manuscript (including the abstract) there is an extensive use of the word "we", which does not fit with the writing of a review article.
According to the suggestion, the term "we" were excluded from the abstract.
Line 12-42
Abstract: (1) Background: Double-strand breaks (DSBs) in a single nucleus are usually measured using the sperm chromatin structure assay (SCSA), sperm chromatin dispersion (SCD) test, and comet assay (CA). Mono-dimensional single-cell pulsed-field gel electrophoresis (1D-SCPFGE) and angle-modulated two-dimensional single-cell pulsed-field gel electrophoresis (2D-SCPFGE) were developed to observe DNA fragmentation in separated motile sperm. (2) Methods: Comparative standards, calibration curves, the required sensitivity, and eligibility criteria for test sperm were set up to validate their measurement principles. (3) Results: The conventional methods overlooked the interference of nucleoproteins in their measurements. In-gel proteolysis improves the measurement accuracies of 1D- and 2D-SCPFGE. Naked DNA is suitable as comparative standards and test specimens. Moreover, several dysfunctions that may induce DNA damage are observed in the separated motile sperm. Overall discussion highlights the need to revisit the conventional univariable analyses based on the SCSA, SCD test, and CA. (4) Conclusions: Human infertility is a complex syndrome, and the aim of quality control in intracytoplasmic sperm injection is to identify the underlying dysfunctions remaining in the separated motile sperm that render them ineligible for injection. Multivariable analyses with special consideration to confounding factors are necessary in future cohort studies.
Simple Summary: Double-strand breaks (DSBs) in a single nucleus are usually measured using the sperm chromatin structure assay (SCSA), sperm chromatin dispersion (SCD) test, and comet assay (CA). Mono-dimensional single-cell pulsed-field gel electrophoresis (1D-SCPFGE) and angle-modulated two-dimensional single-cell pulsed-field gel electrophoresis (2D-SCPFGE) were developed to observe DNA fragmentation in separated motile sperm. Comparative standards, calibration curves, the required sensitivity, and eligibility criteria for test sperm were set up to validate their measurement principles. The conventional methods overlooked the interference of nucleoproteins in their measurements. In-gel proteolysis improves the measurement accuracies of 1D- and 2D-SCPFGE. Naked DNA is suitable as comparative standards and test specimens. Moreover, several dysfunctions that may induce DNA damage are observed in the separated motile sperm. Overall discussion highlights the need to revisit the conventional univariable analyses based on the SCSA, SCD test, and CA. Human infertility is a complex syndrome, and the aim of quality control in intracytoplasmic sperm injection is to identify the underlying dysfunctions remaining in the separated motile sperm that render them ineligible for injection. Multivariable analyses with special consideration to confounding factors are necessary in future cohort studies.
-Infertility is not a symptom. ESHRE's definition of infertility is: “Disease characterized by the failure to establish a clinical pregnancy after 12 months of regular, unprotected sexual intercourse or due to an impairment of a person's capacity to reproduce, either as an individual or with their partner”.
We understand that infertility is a syndrome but not symptom. We distinguished their word senses.
Line 23, 38, 98: syndrome
Line 95, 353, 480, 481: symptom
-I would highly recommend editing on the organization of the manuscript. It starts by describing gold standard protocols for sperm selection such as density gradient and swim-up and than passing to the work that the authors have done. In my opinion the chapter 1 can be deleted from the manuscript.
Line 103-121
Apoptosis increases the apparent density of human sperm due to apoptotic volume decrease, and they lost motility due to membrane damages. We observed that Optiprep sedimentation equilibrium was suitable to separate the immotile sperm with the end stage of DNA fragmentation (DS) from the motile sperm with fibrous DNA (LS). Percoll density gradient centrifugation and the subsequent swim up are used to separate motile sperm in LS. For these reasons, the present method is not popular, we would like to include the method into chapter 1.
-The enumeration is not correct: Chapter 1 is followed by the chapter 2.1 and then proceeds with the chapter 3.
Line 104
2.1. D-SCPFGE is printing error, 2. 1D-SCPFGE is correct.
-The last part of the article is written as an original article in which the authors highlight that they are working on "developing of a modified 2D-SCPGFE 471 and the corresponding calibration standards for the simultaneous observation of early-stage DSBs and SSBs (AP sites)". This is confusing for the reader as I would recommend the authors to explain and highlight once again the pros and the cons of the protocols described in the manuscript.
Line 497-505
According to the suggestion, the following sentence was inserted.
The present review highlights the need to revisit some cohort studies based on CA, SCSA, SCD test, and TUNEL assay. We are developing of a modified 2D-SCPGFE and the corresponding calibration standards for the simultaneous observation of early-stage DSBs and SSBs (AP sites). When the development is complete, we plan to conduct multivariable analyses with the novel method to discuss etiological significance of DNA damages in male infertility. Endogenous and exogenous ROS, the plasma membrane damages, and formation of vacuoles should be taken into account as potential confounding factors, multifaceted approaches are necessary to find answers through the cohort studies.
